# Flexicaulin A, An *ent*-Kaurane Diterpenoid, Activates p21 and Inhibits the Proliferation of Colorectal Carcinoma Cells through a Non-Apoptotic Mechanism

**DOI:** 10.3390/ijms20081917

**Published:** 2019-04-18

**Authors:** Yixuan Xia, Chu Shing Lam, Wanfei Li, Md. Shahid Sarwar, Kanglun Liu, Kwan Ming Lee, Hong-Jie Zhang, Siu Wai Tsang

**Affiliations:** School of Chinese Medicine, Hong Kong Baptist University, Kowloon, Hong Kong SAR, China; 15485005@life.hkbu.edu.hk (Y.X.); 13207504@life.hkbu.edu.hk (C.S.L.); wanfeier@live.cn (W.L.); 15484327@life.hkbu.edu.hk (M.S.S.); liukanglun@hkbu.edu.hk (K.L.); dickyleelol@hotmail.com (K.M.L.)

**Keywords:** cell cycle arrest, colorectal carcinoma, diterpenoid, *ent*-kaurane, p21

## Abstract

Natural products, explicitly medicinal plants, are an important source of inspiration of antitumor drugs, because they contain astounding amounts of small molecules that possess diversifying chemical entities. For instance, *Isodon* (formerly *Rabdosia*), a genus of the Lamiaceae (formerly Labiatae) family, has been reported as a rich source of natural diterpenes. In the current study, we evaluated the in vitro anti-proliferative property of flexicaulin A (FA), an *Isodon* diterpenoid with an *ent*-kaurane structure, in human carcinoma cells, by means of cell viability assay, flow cytometric assessment, quantitative polymerase chain reaction array, Western blotting analysis, and staining experiments. Subsequently, we validated the in vivo antitumor efficacy of FA in a xenograft mouse model of colorectal carcinoma. From our experimental results, FA appears to be a potent antitumor molecule, since it significantly attenuated the proliferation of human colorectal carcinoma cells in vitro and restricted the growth of corresponsive xenograft tumors in vivo without causing any adverse effects. Regarding its molecular mechanism, FA considerably elevated the expression level of p21 and induced cell cycle arrest in the human colorectal carcinoma cells. While executing a non-apoptotic mechanism, we believe the antitumor potential of FA opens up new horizons for the therapy of colorectal malignancy.

## 1. Introduction

Abnormal cell growth gives rise to the formation of tumors, which become either benign or malignant. Without appropriate treatment, malignant tumors, commonly known as cancers, can cause serious illness and death. According to the World Health Organization, cancers are accounted for an estimated 9.6 million deaths worldwide in 2018, and cancers of the colon, lung, breast, prostate, and liver are diagnosed with the highest frequency [1]. In general, tumor cells are defined as highly proliferative because they continue to divide, evading the normal growth inhibitory controls. Apoptosis is one of the major mechanisms that restricts cellular proliferation in response to DNA damage, oncogenic stimuli, and physiological stresses [2,3]. Therefore, it is not surprising that the vast majority of anticancer drugs currently used in clinical oncology are apoptotic inducers. On the other hand, the rate of cellular proliferation is indeed governed by the cell cycle process and rhythm. Each phase of the cell cycle is tightly regulated by receptor collectives, also known as cell-cycle checkpoints [4]. Agents that induce checkpoint arrest may either increase or decrease the quality and rate of cell division; thus, cell cycle arrest has been suggested as another approach to intervene tumor development besides apoptosis [5,6]. In particular, cyclin-dependent kinase inhibitor p21 (CDKN1A or p21) plays an important role in cell cycle regulation [7,8]. According to previous reports, p21 may directly bind to CDK2 or CDK4/6 complexes in triggering cell cycle arrest [9]. Alternatively, p21 may interact with proliferating cell nuclear antigen (PCNA) in modulating cellular proliferation, explicitly in those rapidly dividing cells [10,11]. Taken together, the up-regulation of p21 highly correlates to cell cycle arrest, and thus halts cellular proliferation in various types of cancerous cells [12]. In this regard, p21 induction appears to be beneficial to cancer chemotherapies.

In the quest for chemotherapeutic agents, plant extracts have often served as the starting materials for drug development, since they contain astounding amounts of small molecules, which possess a great variety of core structures with multiple functionalities and unique stereochemical features [13,14]. In recent decades, a number of pharmacological studies had been focused on the administration of diterpenes. Terpenes are a group of hydrocarbons biosynthetically derived from isoprene units, whilst diterpenes are those composed of two isoprene units [15]. If they possess a kaurane skeleton with a configurational inversion at all chirality centers, they are categorized as *ent*-kaurane diterpenes. In the present study, the application of flexicaulin A (FA, molecular weight = 392.5; Figure 1), an *ent*-kaurane diterpenoid abundantly found in *Isodon flexicaulis* H. Hara [16,17], significantly inhibits the proliferation of human colorectal carcinoma cells in vitro and restricts the growth of xenograft tumors in vivo. Regarding its underlying mechanism, FA specifically up-regulated p21 expression and induced cell cycle arrest at G1 and G2. Collectively, our work thereby reveals the antitumor potential of FA and provides a novel insight into how FA inhibits cellular proliferation and induces cell cycle arrest in colorectal carcinoma cells.

## 2. Results

### 2.1. Flexicaulin A Treatment Inhibits Carcinoma Cell Proliferation

In this study, we examined the effect of FA on the cellular proliferation of four human cancer cell lines by means of a sulforhodamine B (SRB) viability assay. The IC_50_ values (i.e., the concentrations required to achieve 50% inhibition over the growth of control cells) of FA in colorectal carcinoma HCT-116, colorectal carcinoma HT-29, melanoma A375, and pancreatic adenocarcinoma PANC-1 cells were 1.32 (3.36), 1.98 (5.05), 2.97 (7.57) and 1.87 μg/mL (4.76 μM), respectively. Based on the IC_50_ values, the human colorectal carcinoma HCT-116 cells were the most sensitive to FA among the four cancer cell lines tested. The anti-proliferative activity of FA against HCT-116 cells was further evaluated by means of a colony formation assay. As shown in Figure 2, FA treatment significantly reduced the clonogenic survival of the adherent cells, while the growth of HCT-116 colonies was almost completely inhibited at 1.95 μg/mL.

### 2.2. Flexicaulin A Treatment Induces Cell Cycle Arrest

To elucidate the underlying anti-proliferative mechanism of FA in human colorectal carcinoma HCT-116 cells, flow cytometric analysis was performed. However, our results demonstrate that no significant apoptotic events were detected in the cells post-FA treatment, even at concentrations higher than its IC_50_ value. In this apoptotic assay, the current mainstay anticancer agent paclitaxel (PTX) served as a positive reference. When PTX was applied at 10 ng/mL, a concentration around its IC_50_ value against HCT-116 cells, remarkable apoptosis was perceived. Representative analytic plots of apoptosis are shown in Figure 3a, whereas the percentages of apoptotic cells upon incubation with different compounds are provided in Figure 3b. In regard to these data, we speculated that the anti-proliferative effect of FA against HCT-116 cells was plausibly derived from a non-apoptotic mechanism. After analyzing the DNA content of the cancer cells, we observed that FA treatment at concentrations lower than 1 μg/mL caused a notable increase of cell number in the G1 fraction (72.72% ± 2.94) in relation to the DMSO-treated control. When FA was applied at higher concentrations, e.g., 1.25 and 1.5 μg/mL, a significant arrest at G2 was then obtained (Figure 4). In this regard, FA appears to be a modulating agent of cell cycle distribution in colorectal carcinoma cells. As senescence has been recognized as an irreversible cell cycle arrest, particularly in the G1 phase, we thus examined the inducing effect of FA on senescence-associated β-galactosidase (SA-β-gal) activity. Upon the supplying of the enzyme substrate X-gal, considerable senescent signals (cyan) were observed in the HCT-116 cells, only when the concentration of FA was increased to 1.5 μg/mL, a concentration that exceeds its IC_50_ value and causes significant cell death (Figure 5). Thus, the anti-proliferative effect of FA on HCT-116 cells was not primarily derived from its induction of senescence. In this SA-β-gal staining experiment, 5-fluorouracil (5-FU, 0.25 μg/mL) at a concentration much below its IC_50_ value served as a positive reference. Indeed, 5-FU is the most commonly used chemotherapeutic for colorectal carcinoma that induces cellular senescence. In addition to cyan development, the senescent cells also exerted an altered morphology, e.g., flattened and enlarged. On the other hand, we also performed a quantitative polymerase chain reaction (qPCR) array, profiling for the expression levels of a panel of senescence-related genes as a parallel experiment to SA-β-gal staining. Among the 84 test genes, only four of them were affected by the FA treatment (1 μg/mL) with statistical significance (*p* < 0.05; Figure 6). Taking the above results together, we conclude that the anti-proliferative property of FA in colorectal carcinoma cells was principally derived from cell cycle arrest, rather than apoptosis or senescence.

### 2.3. Flexicaulin A Targets the p53/p21 Signaling Pathway

From the qPCR array data, we noticed that CDKN1A (p21) was elevated 2.3-fold upon the treatment with FA at 1 μg/mL (Table 1). As such, the up- and down-stream regulators of p21 might also be considerably altered by FA. From the immunoblots, we observed that the protein levels of p53 and p21 in HCT-116 cells increased concentration-dependently post-FA treatment (Figure 7a). Such a result implies that FA activates the p53/p21 axis. As a consequence, the downstream targets of p21, such as p16, p27, RB, and E2F1, are stabilized in the form of a repressor complex, which suppresses the activity of cyclin D1 and arrests cell cycle progression. The immunofluorescent images further reveal up-regulated p21 nuclear expression by FA treatment in the HCT-116 cells (Figure 7b). Collectively, our results suggest that the antitumor effect of FA is majorly associated with the p21-mediated execution of cell cycle arrest.

### 2.4. Flexicaulin A Treatment Reduces Tumor Development in A Colorectal Carcinoma Xenograft Model

To validate the antitumor efficacy of FA in vivo, FA was administered to the HCT-116 xenograft tumor-bearing nude mice in a 14-day trial. At the end of the experimental trial, we found that FA treatments at 20 and 40 mg/kg notably reduced the growth of the colorectal carcinoma xenografts (Figure 8), in terms of tumor volume (Figure 9a) and tumor weight (Figure 9b), with a statistical significance when compared to the vehicle group (*n* = 10/group). Although the tumor-suppressive effect of FA was not comparable to that of PTX, which is a mainstay chemotherapeutic agent for malignancies, the toxicity of FA (20 or 40 mg/kg) appeared to be minimal, as no marked effect on body weight loss (Figure 10a) or other adverse symptoms were observed in the FA-treated mice. On the contrary, the administration of PTX (10 mg/kg) caused 20% animal death during the experimental period (Figure 10b).

## 3. Discussion

In the present study, we elucidated the anti-proliferative property of FA and its induction of cell cycle arrest using cellular platforms, and validated its in vivo antitumor efficacy in an athymic, xenografted mouse model of colorectal carcinoma. In general, a large number of anticancer agents on the market nowadays are apoptotic inducers, since apoptosis is an acknowledged mechanism to eliminate damaged or pre-neoplastic cells, thus restricting tumorigenesis [18,19]; however, malignant cells often escape from pro-apoptotic signals and retain their replicative capability [20]. As far as we are concerned, the rapid development of resistance to apoptosis largely hampers the usefulness and effectiveness of many conventional anticancer chemotherapeutics [21]. If cell cycle arrest serves as a tumor suppressive mechanism alternative to apoptosis, the use of cell cycle modulators will be beneficial to the clinical management of malignancies [22]. The data of our flow cytometric analysis shows that the anti-proliferative effect of FA on human colorectal carcinoma cells is primarily derived from a non-apoptotic mechanism (Figure 3), as it induces cell cycle arrest (Figure 4).

In regard to the molecular mechanism, the cell cycle arresting effect of FA on human colorectal carcinoma HCT-116 cells was plausibly established and maintained by engaging the p53/p21 signal transduction pathway. FA appears to not be a potent inducing agent of DNA damage, but it up-regulated the tumor suppressor p53 at the translational level, as well as its downstream target p21 (Figure 7). Interestingly, the human colorectal carcinoma HT-29 cells were less sensitive to FA when compared to the HCT-116 cells; hence, we suggest that p53 may play a role in the molecular action of FA. According to previous findings, HT-29 cells express a mutant form of p53, while the R273H mutation was reported to be non-functional for the traditional p53 activities, such as DNA binding [23]. In this regard, we believe the countering effect of FA against the unlimited replicative potential of colorectal carcinoma cells became more potent when p21 activation was orchestrated by up-regulation of p53 [24]. In accordance with other studies, p21 induction can be modulated via p53-dependent or p53-independent mechanisms in rapidly proliferating cancer cells [7,25]. Nevertheless, the relative preponderance of the p53/p21 pathway in anticancer mechanisms varies among cell types, as well as species.

For the development of novel anticancer agents, medicinal plants are undeniably the infinite resources of chemical entities. The bioactivities of diterpenoids, particularly those possessing an *ent*-kaurane structure, have been gaining increasingly great attention, since they have been shown to attenuate autoimmune inflammation via inhibiting nuclear factor-kappaB (NF-κB) activation [26]. Apart from the inflammatory cascade, NF-κB signaling is indeed involved in a number of other mammalian cellular processes, including tumorigenesis [27]. Therefore, we suggest *ent*-kaurane diterpenoids are potential anticancer remedies. According to previous studies, *Isodon* (formerly *Rabdosia*), a genus of the Lamiaceae (formerly Labiatae) family comprising 150 species, has been reported as a rich source of natural *ent*-kaurane diterpenes [15,17]. In our present work, FA, an *ent*-kaurane diterpenoid richly found in *I. flexicaulis*, provided promising anti-proliferative activity against carcinoma cells—HCT-116 in particular. In addition to the attenuated NF-κB activation, FA increased the expression levels of p21 and p53 (Figure 7) and caused significant cell cycle arrest (Figure 4). As modulators of cell cycle have been demonstrated as potent anti-proliferative agents, they are plausible components of chemotherapies for cancers [22]. As far as we know, many cytotoxic agents may deliver adverse effects, although they exert a remarkable effect on tumor reduction in vivo. In Nassier’s study, the therapeutic activity of PTX (10 mg/kg, i.p.) led to a substantial loss of animals suffering from protein deficiency or malnutrition, and a survival rate of approximately 20% was attained at the end of the one-month experimental trial [28]. In our present study, PTX treatment also caused significant animal death during the course of our experiment (Figure 10b), even the administration period was shortened to 14 days. When PTX was given at lower dosages, its reducing effect on tumor growth largely declined. These results vastly implicate the undesirable toxicity and narrow therapeutic index of PTX, though it is one of the most commonly used anticancer agents in clinical oncology nowadays. Contrarily, the high-dose administration of FA (i.e., 40 mg/kg) did not cause detrimental consequences in nude mice in our current study. While exhibiting an encouraging in vivo antitumor effect, we believe FA is a safe chemotherapeutic agent for tumorous pathologies and deserves detailed investigations on its pharmacokinetic and metabolomic characteristics. With a flexible and manipulatable structure, FA can serve as an ideal synthetic scaffold for the generation of potent anticancer analogs. Structural modification of its functional groups or active sites may assist in achieving optimal anti-proliferative activity or drug selectivity towards cancer cells, and make FA more clinically imminent.

## 4. Materials and Methods

### 4.1. Materials

The *ent*-kaurane diterpenoid FA was purchased from Gihon Biotech, Hong Kong, and dissolved in dimethylsulfoxide (DMSO) to a stock solution at 1 mM, which was kept at –20 °C until use. The positive antitumor references 5-FU and PTX were respectively purchased from Abcam, Cambridge, UK and Santa Cruz Biotechnology, Dallas, TX, United States.

### 4.2. Cell lines and Culture Condition

Human colorectal carcinoma cell lines HCT-116 and HT-29, human melanoma cell line A375, and human pancreatic ductal adenocarcinoma cell line PANC-1 were purchased from American Type Culture Collection (ATCC) and maintained in McCoy’s 5A (Gibco) or DMEM (Gibco), supplemented with 10% fetal bovine serum (Gibco), 1% penicillin-streptomycin (Gibco) in a 5% CO_2_, and 95% air-humidified atmosphere at 37 °C. Cells with fewer than 30 passages were used in our experiments.

### 4.3. Cell Viability Assay

The cytotoxicity of FA was evaluated in terms of cellular protein content, according to an established protocol [29]. Carcinoma cells were seeded in 96-well plates at a density of 8 × 10^3^ cells/well, and incubated with FA (0 to 10 μg/mL) for 24 to 72 hours (h). At the end of incubation, cells were treated with 50% trichloroacetic acid at 4 °C for at least 30 min for protein fixing, and were then washed with water four times. Upon completely dried, 100 μL of 0.4% SRB in 1% acetic acid was added to each well. After 30 min of incubation at room temperature, cells were washed with 1% acetic acid four times. Fixed proteins were fully solubilized with 10 mM Tris base solution (pH 10, 200 μL/well). The spectrophotometric absorbance of the samples was measured at 515 nm, using a microplate reader (Bio-Rad, Hercules, CA, USA).

### 4.4. Colony Formation Assay

HCT-116 cells were cultured in 2 mL of McCoy’s 5A medium in six-well plates (600 cells/well) and treated with FA (0.65 to 1.95 μg/mL) or 0.5% DMSO for 14 days. At the time of harvest, the cells were fixed in 4% paraformaldehyde and subjected to crystal violet staining prior to image capture with a light microscope (Leica) [29].

### 4.5. Flow Cytometric Analyses

Cells were seeded in six-well plates at a density of 1 × 10^6^ cells/well, and were incubated with serial dilutions of FA (0.65 to 2.64 μg/mL), PTX (10 ng/mL), or 0.5% DMSO for 48 h. After centrifugation, cells were re-suspended in 500 μL of binding buffer containing 5 μL of propidium iodide (PI) and 5 μL of annexin V-FITC, and then incubated for 15 min in the dark, according to the manufacturer’s instructions (Thermo Fisher Scientific, Eugene, OR, USA). In another set of experiments, cells were subjected to ribonuclease treatment prior to PI staining for the analysis of cell cycle phases. Signals of the cell suspension were analyzed using a flow cytometer (BD, San Jose, CA, USA).

### 4.6. Senescence-Associated β-Galactosidase (SA-β-gal) Colorimetric Assay

HCT-116 cells were seeded in 12-well plates at a density of 1 × 10^5^ cells/well, and incubated with serial dilutions of FA (0.5 to 1.5 μg/mL), 5-FU (0.25 μg/mL), or 0.5% DMSO for 48 h. Staining for SA-β-gal activity was performed using a commercial kit (Cell Signaling Technology, Danvers, MA, USA ), according to the manufacturer’s instructions, in which X-gal (pH 6.0) was used as a substrate for the enzyme. Images were taken using the AF6000 inverted microscope (Leica, Mannheim, Germany).

### 4.7. Quantitative Polymerase Chain Reaction (qPCR) Arrays

Total RNA was extracted from HCT-116 cells using the RNeasy Mini Kit (Qiagen, Valencia, CA, USA) according to the manufacturer’s instructions. One μg of the total RNA of each sample was transcribed into cDNA using PrimeScript RT master mix (Takara) in a total volume of 20 μL. The synthesized cDNA samples were mixed with SsoFast EvaGreen PCR Master Mix (Bio-Rad), and applied to the RT^2^ Profiler PCR Arrays (Qiagen, 96-well plate format). The real-time PCR program was executed in the CFX connect system (Bio-Rad). Expression of the gene of interest of each sample was normalized to the housekeeping genes *ACTB*, *B2M*, *GAPDH*, *HPRT1*, and *RPLP0* while their fold changes were calculated using the software provided by Qiagen that compiled using the comparative CT (2^−ΔΔ*C*T^) method.

### 4.8. Western Blot Analysis

Proteins were extracted from HCT-116 cells using an ice-cold lysis buffer containing protease and phosphatase inhibitors (Thermo Fisher Scientific). Cell lysates were loaded, separated by 10% to 15% sodium dodecyl sulfate (SDS)-polyacrylamide gel electrophoresis, and transferred onto polyvinylidene fluoride (PVDF) membranes (Western Bright) by wet electroblotting. The membranes were blocked with 5% non-fat dry milk in Tris-buffered saline containing 0.1% Tween 20, for 1 h at room temperature, then were probed with a primary antibody against p21 (Abcam), p53 (Cell Signaling Technology), p16^INK4a^ (Thermo Fisher Scientific), p27^KIP1^ (Abcam), pRB (Abcam), E2F1 (Abcam), or GAPDH (Bio-Rad) overnight at 4 °C; the membranes were subsequently incubated with corresponsive horseradish peroxidase-conjugated secondary antibodies (Santa Cruz Biotechnology). The proteins were eventually visualized by utilization of an enhanced chemiluminescence kit (Western Bright).

### 4.9. Immunofluorescent Staining

HCT-116 cells were seeded (1 × 10^5^ cells/mL) in eight-well chamber slides, and treated with FA (1.32 μg/mL) for 48 h. When harvested, the cells were washed with phosphate buffer saline and fixed in ice-cold acetone:methanol (1:1, *v*/*v*) for 20 min. After rinsing, fixed cells were blocked with 3% bovine serum albumin and probed with a primary antibody against p21 (Abcam). Subsequently, cells were incubated with an anti-rabbit, FITC-conjugated secondary antibody prior to mounting, using a mounting medium containing 4′, 6-diamidino-2-phenylindole (DAPI; Sigma-Aldrich). Images were captured and analyzed using a Nikon microscope equipped with SPOT Advanced software (Version 4.6, Sterling Heights, MI, USA).

### 4.10. Colorectal Carcinoma Xenograft Model

The experimental procedures in the present study were approved by the Committee on the Use of Human and Animal Subjects in Teaching and Research (HASC) of Hong Kong Baptist University (HASC/17-18/0833, March, 2017), Hong Kong SAR, China. Male BALB/c nude mice, specific pathogen free (SPF) class, aged 6 to 7 weeks old, were purchased from BioLASCO, Taipei, Taiwan. According to our previously reported protocol [30], HCT-116 cells (1 × 10^6^ in 100 μL) were implanted subcutaneously in the right and left flanks of nude mice for tumor development. Tumor size was monitored by measuring two perpendicular diameters with a caliper every other day. When palpable tumors were formed (~100 mm^3^) seven days after implantation, the xenograft tumor-bearing mice were assigned into four groups (*n* = 10/group), which were (i) vehicle, (ii) FA 20 mg/kg, (iii) FA 40 mg/kg, and (iv) PTX 10 mg/kg. The FA treatment groups were intraperitoneally (i.p.) administered with FA at 20 or 40 mg/kg every other day for a total of 14 days, while the vehicle group merely received the 2.5% EtOH and Cremphor EL solution, which was the dissolving solution for our test compounds. To the PTX treatment group, PTX was given at 10 mg/kg (i.p.), and used as a positive reference agent. The tumor volume and body weight of mice were recorded at indicated time points during the course of experiment. Tumor volume was calculated as volume = length × width^2^ × 0.5. The weight of the tumor xenograft was assessed at the time of sacrifice.

### 4.11. Statistical Analysis

The statistical differences among experimental groups were determined using one-way analysis of variance (ANOVA), followed by Tukey’s test as a post-hoc test. All values are expressed as means ± standard deviation (SD). A *p* value of <0.05 is accepted as statistically significant.

## 5. Conclusions

The present study demonstrates that FA attenuated the proliferation of human colorectal carcinoma cells via the induction of p21-mediated cell cycle arrest. Importantly, the in vivo inhibitory effect of FA against the growth of colorectal carcinoma xenograft is encouraging. While executing a non-apoptotic mechanism, we believe the antitumor potential of FA opens up new horizons for the therapy of colorectal malignancy.

## Figures and Tables

**Figure 1 ijms-20-01917-f001:**
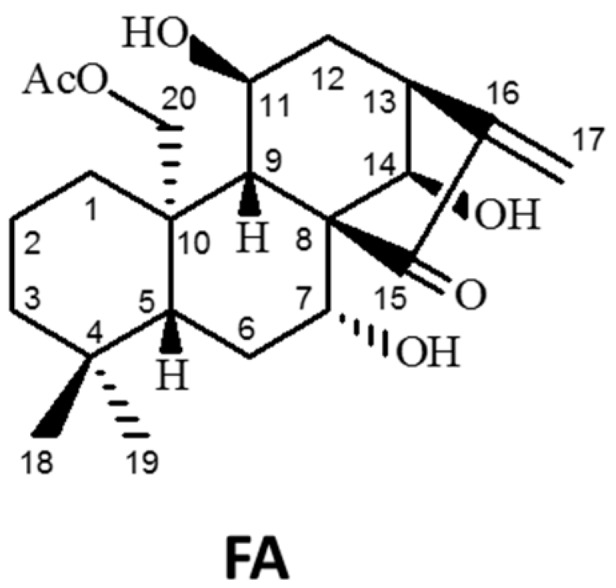
The chemical structure of *ent*-kaurane diterpenoid flexicaulin A (FA).

**Figure 2 ijms-20-01917-f002:**
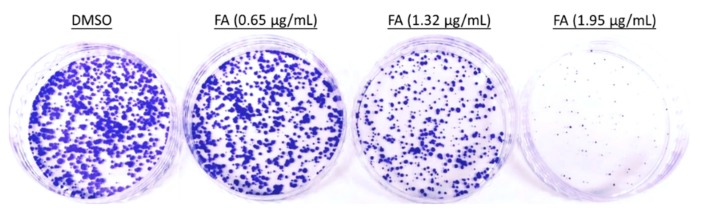
The inhibitory effect of FA on colony formation capacity in HCT-116 cells. In 6-well plates, clonogenic assay was performed post treatment with dimethylsulfoxide (DMSO) (0.5%) or FA at the indicated concentrations for 14 days, and cells in each well were stained with crystal violet at the end of the experiment. Shown are representative images from at least three independent experiments.

**Figure 3 ijms-20-01917-f003:**
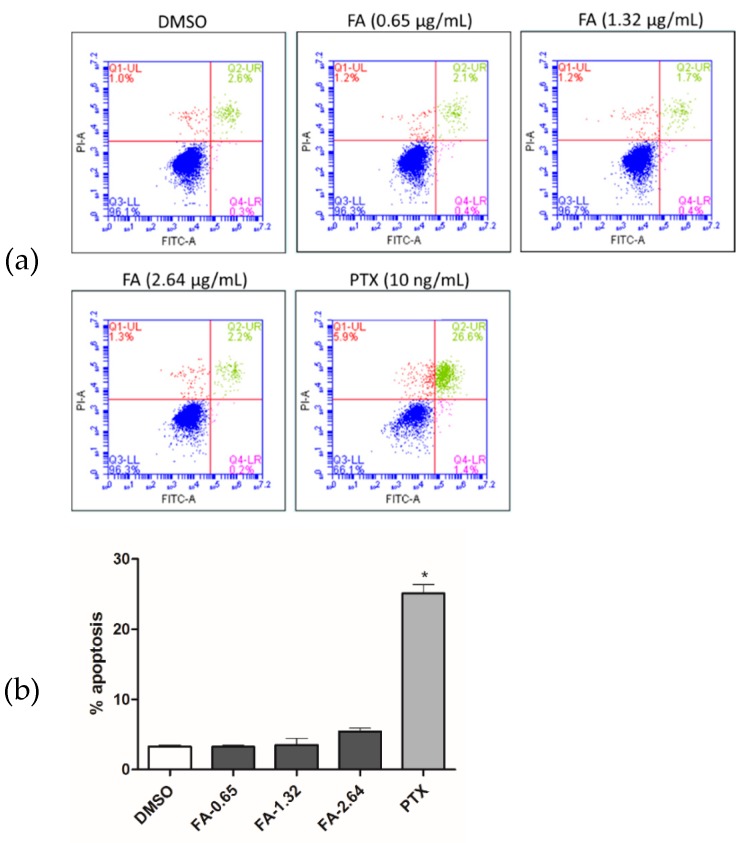
Flow cytometric assessment of apoptosis. HCT-116 cells were treated with DMSO (0.5%), FA (0.65, 1.32 or 2.64 μg/mL), or paclitaxel (PTX) (10 ng/mL) for 48 h prior to labeling with propidium iodide (PI) and fluorescein isothiocyanate (FITC)-annexin V. The Q1-UL (red) quadrant shows the percentage of necrotic cells, the Q2-UR (green) and Q4-LR (magenta) quadrants represent late and early apoptotic events, respectively, while the Q3-LL (blue) quadrant denotes the population of viable cells. (**a**) Shown here are the representative analytic plots of four independent experiments. (**b**) Apoptotic data (i.e., Q2-UR + Q4-LR) in the bar chart are expressed as mean ± standard deviation (SD) of four independent experiments (* *p* < 0.001 when compared to DMSO control).

**Figure 4 ijms-20-01917-f004:**
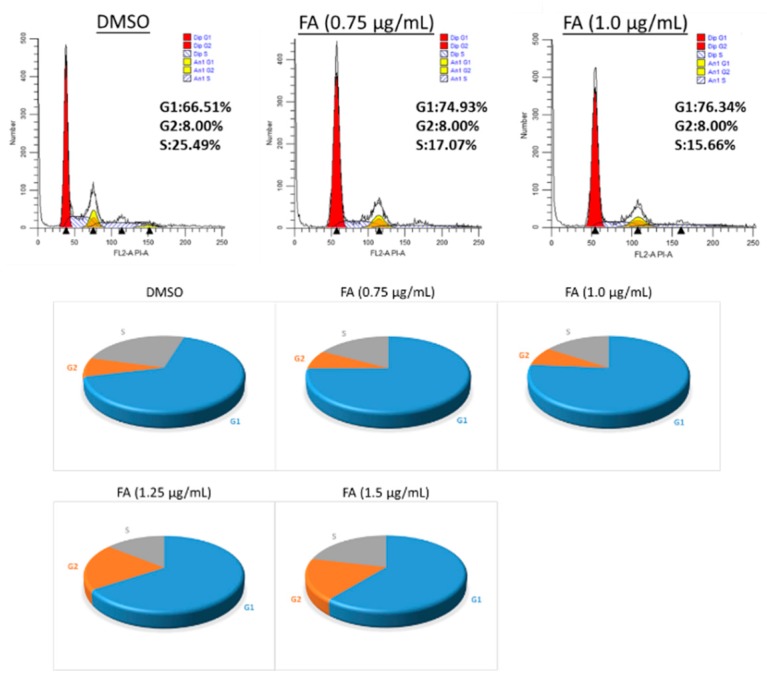
Evaluation of cell cycle distribution. HCT-116 cells were treated with DMSO (0.5%) or FA (0.75–1.5 μg/mL) for 12 h prior to flow cytometric analysis. Shown here are representative distributions from three independent trials. In the pie charts, grey sections denote the S phase, blue sections denote the G1 phase, and orange sections denote the G2 phase.

**Figure 5 ijms-20-01917-f005:**
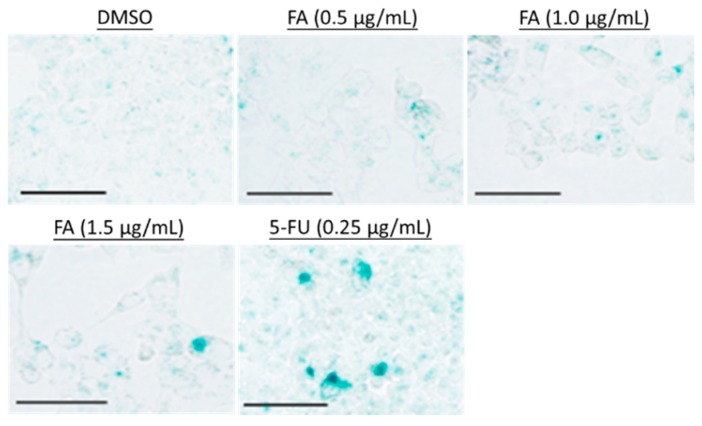
Evaluation of senescent status. HCT-116 cells were treated with DMSO (0.5%), FA (0.5–1.5 μg/mL), or 5-fluorouracil (5-FU) (0.25 μg/mL) for 48 h prior to staining. The cells showing SA-β-gal activity were stained cyan upon the addition of the substrate X-gal. Shown here are representative images from at least three independent batches of staining (scale bar = 50 μm).

**Figure 6 ijms-20-01917-f006:**
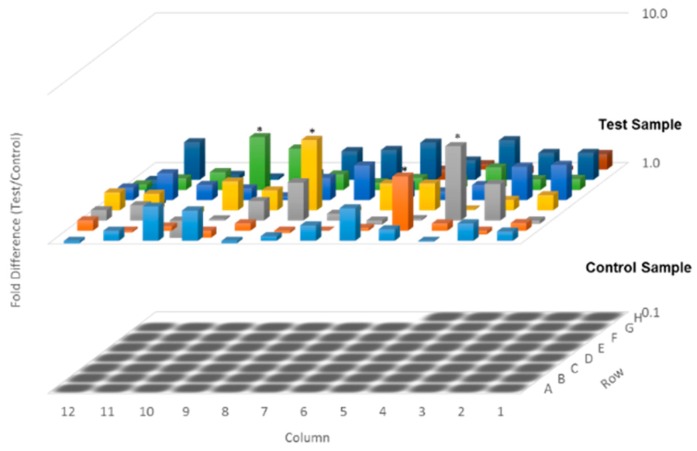
Assessment of senescence-related regulators. HCT-116 cells were treated with DMSO (0.5%) or FA at 1 μg/mL for 48 h, prior to mRNA extraction for a quantitative polymerase chain reaction (qPCR) array; data are calculated from three independent experiments (* *p* < 0.05 when compared to DMSO control).

**Figure 7 ijms-20-01917-f007:**
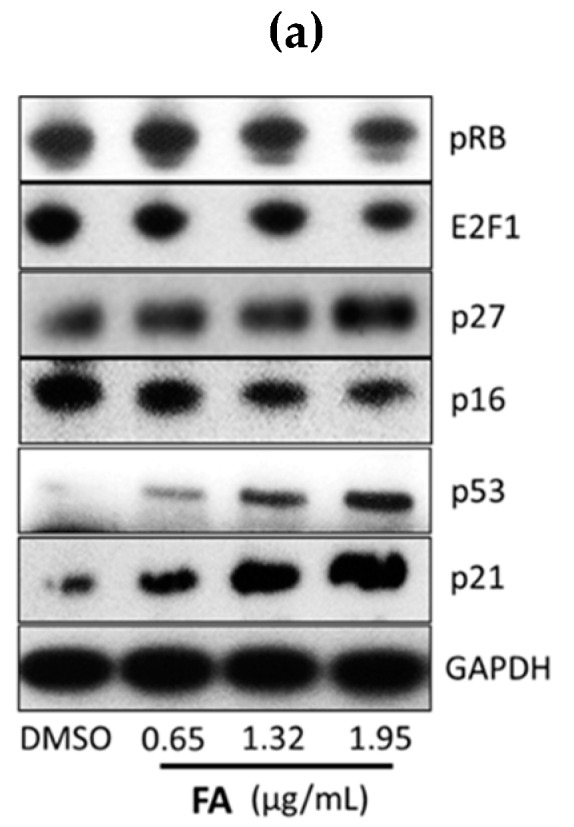
The up-regulation of p21 and related regulators post-FA treatment. (**a**) For Western blotting analysis, cells were treated with DMSO (0.5%) or FA at indicated concentrations for 48 h prior to protein extraction. GAPDH and histone H3 were served as loading references of the cytoplasmic and nuclear fractions, respectively. Shown here are representative immunoblots from at least three independent experiments. (**b**) For immunofluorescent staining, cells were treated with DMSO (0.5%) or FA (1.32 μg/mL) for 48 h prior to fixing. The immunoreactivities of p21 were stained green with FITC, whereas nuclei were stained blue with 4′,6-diamidino-2-phenylindole (DAPI). Shown here are representative immunofluorescent images from three independent experiments (scale bar = 100 μm).

**Figure 8 ijms-20-01917-f008:**
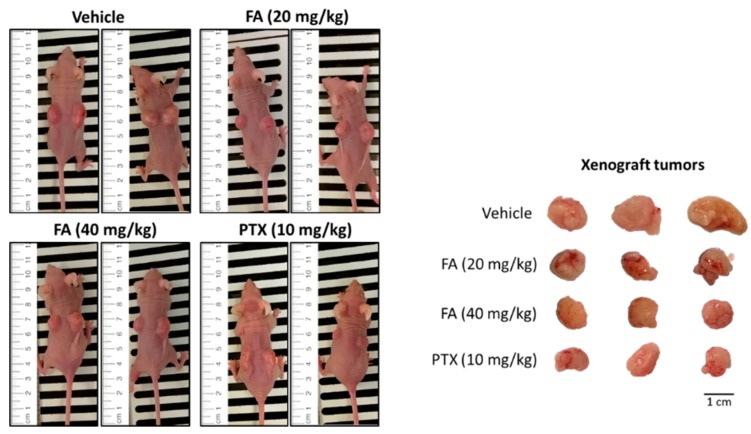
The antitumor efficacy of FA in HCT-116 xenograft-bearing nude mice. When the xenograft tumors reached about 100 mm^3^, vehicle solution, FA (20 or 40 mg/kg), or PTX (10 mg/kg) was intraperitoneally administered to the tumor-bearing mice every other day for 14 days. The subcutaneous tumors were excised at the end of the animal trial.

**Figure 9 ijms-20-01917-f009:**
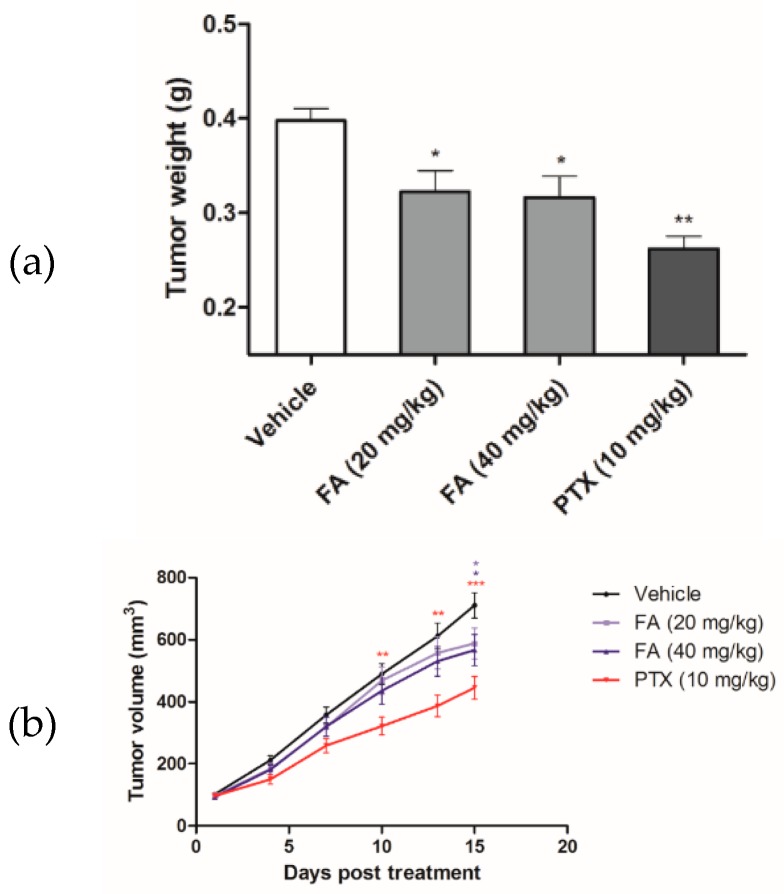
The antitumor effect of FA was assessed in terms of (**a**) weight of excised tumors (* *p* < 0.05 and ** *p* < 0.01 when compared to the vehicle group), and (**b**) volume of the growing subcutaneous tumors (* *p* < 0.05, ** *p* < 0.01, and *** *p* < 0.001 when compared to the vehicle group).

**Figure 10 ijms-20-01917-f010:**
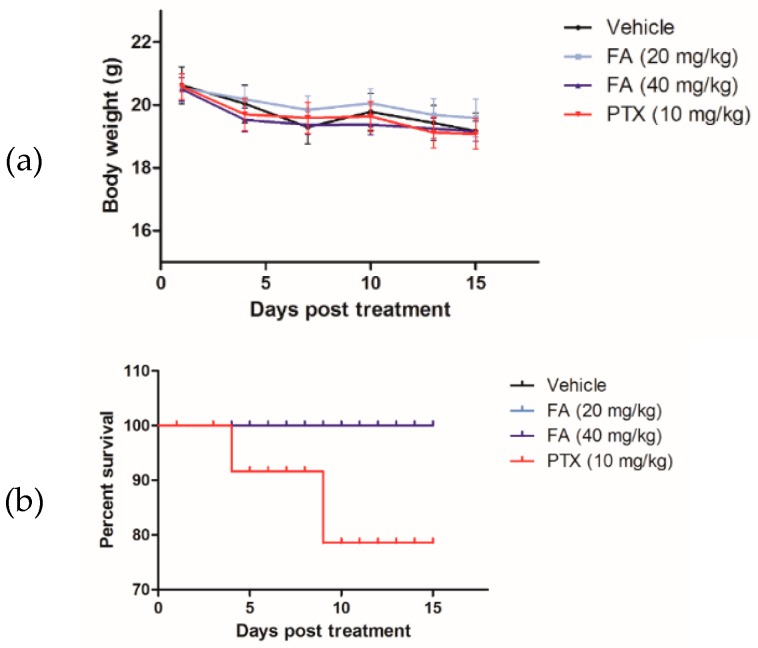
The toxicity of FA and PTX was assessed in terms of (**a**) the body weight of mice and (**b**) the survival rate of animals during the course of experimental period.

**Table 1 ijms-20-01917-t001:** The qPCR array profiling of senescence-related genes in HCT-116 cells upon FA treatment (1 μg/mL). Red indicates changes greater than 2 folds.

Symbol	Well	Fold Change
		Test Sample/Control Sample
ABL1	A01	1.157
AKT1	A02	1.309
ALDH1A3	A03	0.989
ATM	A04	1.199
BMI1	A05	1.657
CALR	A06	1.270
CCNA2	A07	1.076
CCNB1	A08	0.966
CCND1	A09	1.597
CCNE1	A10	1.696
CD44	A11	1.169
CDC25C	A12	0.966
CDK2	B01	1.124
CDK4	B02	0.946
CDK6	B03	1.117
CDKN1A	B04	**2.305**
CDKN1B	B05	1.045
CDKN1C	B06	1.001
CDKN2A	B07	0.965
CDKN2B	B08	1.119
CDKN2C	B09	0.898
CDKN2D	B10	1.068
CHEK1	B11	0.975
CHEK2	B12	1.173
CITED2	C01	0.950
COL1A1	C02	1.754
COL3A1	C03	**3.141**
CREG1	C04	1.024
E2F1	C05	0.940
E2F3	C06	1.110
EGR1	C07	1.797
ETS1	C08	1.353
ETS2	C09	1.019
FN1	C10	0.766
GADD45A	C11	1.265
GLB1	C12	1.172
GSK3B	D01	1.262
HRAS	D02	1.172
ID1	D03	1.013
IFNG	D04	1.514
IGF1	D05	1.514
IGF1R	D06	0.950
IGFBP3	D07	**2.957**
IGFBP5	D08	1.361
IGFBP7	D09	1.561
ING1	D10	0.931
IRF3	D11	1.291
IRF5	D12	1.316
IRF7	E01	1.721
MAP2K1	E02	1.675
MAP2K3	E03	1.262
MAP2K6	E04	1.144
MAPK14	E05	1.264
MDM2	E06	1.710
MORC3	E07	1.409
MYC	E08	1.059
NBN	E09	1.199
NFKB1	E10	1.266
NOX4	E11	1.514
PCNA	E12	1.216
PIK3CA	F01	1.185
PLAU	F02	1.181
PRKCD	F03	1.416
PTEN	F04	1.179
RB1	F05	1.205
RBL1	F06	1.122
RBL2	F07	1.271
SERPINB2	F08	1.893
SERPINE1	F09	**2.258**
SIRT1	F10	1.317
SOD1	F11	1.193
SOD2	F12	1.089
SPARC	G01	1.514
TBX2	G02	1.514
TBX3	G03	1.841
TERF2	G04	1.344
TERT	G05	1.772
TGFB1	G06	1.581
TGFB1I1	G07	1.553
THBS1	G08	1.104
TP53	G09	1.036
TP53BP1	G10	1.075
TWIST1	G11	1.785
VIM	G12	0.662
ACTB	H01	1.069
B2M	H02	0.945
GAPDH	H03	0.900
HPRT1	H04	1.085
RPLP0	H05	0.955

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
