# Peer review of "Flexicaulin A, An ent-Kaurane Diterpenoid, Activates p21 and Inhibits the Proliferation of Colorectal Carcinoma Cells through a Non-Apoptotic Mechanism"

_ijms, 2019, doi:10.3390/ijms20081917_

Round 1
Reviewer 1 Report
The paper reports over antiproliferative activity of flexicaulin A (FA) towards CRC cells. Authors demonstrated that the antiproliferative effects occur via cell cycle arrest and not via cell death.
The paper is interesting but might be improved by authors addressing the following points:
1) what was the rationale to use the selected tumor cel lines - CRC. pancreas, melanoma?
2) authors demonstrate cell cycle cycle arrest and increased expression of p21 while arguing against the presence of apoptosis and senescence. What si then the expected ultimate outcome of the treatment? Cells do not die but are arrested. Permanently or only temporarily? If only temporarily, they may return to the cycle and continue which would not be the anticipated reusult.
3) Did authors check possibly other types of cell death beyond apoptosis?
4) Authors speculated about the role of p53 in sensitivity to FA but do not bring the evidence beyond demonstration of p53 expression in HCT-116 cells. Here the similar expression pattern should be provided in case of HT-29 cells to compare the expresison of p53 following the tretament with FA directly.
5) Fig. 7A nuclear fraction - what is meant to demonstrate?
6) Fig. 7B - scales are missing, magnification
7) Fig. 2 statistical analysis is missing
8) in section 2.2 authors state that " cells were enrlarged and flattened yet provide not data for it.
9) Fig. 6 - not optimal form of graph, change it.
Author Response
Thanks for Reviewer 1's comment, we have the following responses:
1) what was the rationale to use the selected tumor cel lines - CRC. pancreas, melanoma?
Response: Colorectal cancer, pancreatic cancer and melanoma are the most common cancers worldwide. Thus, our lab chose to screen for potential agents against the corresponding cancer cell lines.
2) authors demonstrate cell cycle cycle arrest and increased expression of p21 while arguing against the presence of apoptosis and senescence. What si then the expected ultimate outcome of the treatment? Cells do not die but are arrested. Permanently or only temporarily? If only temporarily, they may return to the cycle and continue which would not be the anticipated reusult.
Response: The outcome appears not a temporary one. A significant decline of cell proliferation was well observed upon the incubation of flexicaulin A from 24-96 hours in the CRC cells in our present study. We had performed our experiments on apoptosis, senescence and cell cycle distribution at several time points. As reported by others (e.g. Correia et al. Front Microbiol 2017), cells may return to the cycle upon certain specific stimuli, which have not been tested in the current study.
3) Did authors check possibly other types of cell death beyond apoptosis?
Response: Besides apoptotic/necrotic events, we had also checked the induction of flexicaulin A on autophagy; however, the result was negative as none of the autophagic markers, namely BECLIN 1, LC3-I and LC3-II, was significantly altered.
4) Authors speculated about the role of p53 in sensitivity to FA but do not bring the evidence beyond demonstration of p53 expression in HCT-116 cells. Here the similar expression pattern should be provided in case of HT-29 cells to compare the expresison of p53 following the tretament with FA directly.
Response: In this study, we primarily focused on the effect of FA on HCT-116 cells. In our preliminary screening assay, we found that the sensitivity of HT-29 cells to FA was lower as shown by a higher IC50 value. That's why we speculated about the involvement of p53. We will continue on our investigation in this direction, and will gather more data for our next manuscript.
5) Fig. 7A nuclear fraction - what is meant to demonstrate?
Response: We had discussed about the inactivation of NF-kB in our manuscript (section 3, lines 214-215), which was in agreement to some previous reports. However, we have deleted the nuclear fraction in the revised manuscript according to the reviewer's suggestion.
6) Fig. 7B - scales are missing, magnification
Response: A scale bar (100 um) was clearly presented in each of the micrograph in Fig. 7B. and was described in the figure legend.
7) Fig. 2 statistical analysis is missing
Response: The decline of number of HCT-116 colonies was very obvious in Fig. 2, where the statistical analysis appears a surplus.
8) in section 2.2 authors state that " cells were enrlarged and flattened yet provide not data for it.
Response: In Fig. 5, the cells stained cyan were enlarged and flattened (e.g. those treated with FA or 5-FU) when comparing to the DMSO-treated control cells.
9) Fig. 6 - not optimal form of graph, change it.
Response: The graph in Fig. 6 was made using the automatic software provided by Qiagen, which is the manufacturer of the qPCR arrays. It's hard for us to request a form change from them. However, Table 1 serves as a detailed file supplementary to Fig. 6.

Reviewer 2 Report
This work is well structured and amply demonstrates in all the experiments conducted that Flexicaulin A has antiproliferative activity on colon cancer cells. In nature there are an infinite number of resources, including plants, which are rich in natural active molecules useful for preventing, combating and curing cancer. One thing I disagree with the authors when they conclude that FA could be an alternative to chemotherapy. I will conclude by saying that given the interesting results obtained in this study and that they should be expanded with other research, new horizons could open up for the therapy of the colon cancer.
Author Response
This work is well structured and amply demonstrates in all the experiments conducted that Flexicaulin A has antiproliferative activity on colon cancer cells. In nature there are an infinite number of resources, including plants, which are rich in natural active molecules useful for preventing, combating and curing cancer. One thing I disagree with the authors when they conclude that FA could be an alternative to chemotherapy. I will conclude by saying that given the interesting results obtained in this study and that they should be expanded with other research, new horizons could open up for the therapy of the colon cancer.
Response: Thanks for Reviewer 2's comment, we have revised our concluding sentences in the abstract (lines 25-27) and the "Conclusions" section (lines 239-241).
